

# Gene-based association study for lipid traits in diverse cohorts implicates *BACE1* and *SIDT2* regulation in triglyceride levels

Angela Andaleon[1,2], Lauren S. Mogil[1] and Heather E. Wheeler[1,2,3,4]

[1] Department of Biology, Loyola University Chicago, Chicago, IL, United States of America
[2] Program in Bioinformatics, Loyola University Chicago, Chicago, IL, United States of America
[3] Department of Computer Science, Loyola University Chicago, Chicago, IL, United States of America
[4] Department of Public Health Sciences, Loyola University Chicago, Maywood, IL, United States of America

## ABSTRACT

Plasma lipid levels are risk factors for cardiovascular disease, a leading cause of death worldwide. While many studies have been conducted on lipid genetics, they mainly focus on Europeans and thus their transferability to diverse populations is unclear. We performed SNP- and gene-level genome-wide association studies (GWAS) of four lipid traits in cohorts from Nigeria and the Philippines and compared them to the results of larger, predominantly European meta-analyses. Two previously implicated loci met genome-wide significance in our SNP-level GWAS in the Nigerian cohort, rs34065661 in *CETP* associated with HDL cholesterol ($P = 9.0 \times 10^{-10}$) and rs1065853 upstream of *APOE* associated with LDL cholesterol ($P = 6.6 \times 10^{-9}$). The top SNP in the Filipino cohort associated with triglyceride levels (rs662799; $P = 2.7 \times 10^{-16}$) and has been previously implicated in other East Asian studies. While this SNP is located directly upstream of well known *APOA5*, we show it may also be involved in the regulation of *BACE1* and *SIDT2*. Our gene-based association analysis, PrediXcan, revealed decreased expression of *BACE1* and decreased expression of *SIDT2* in several tissues, all driven by rs662799, significantly associate with increased triglyceride levels in Filipinos (FDR < 0.1). In addition, our PrediXcan analysis implicated gene regulation as the mechanism underlying the associations of many other previously discovered lipid loci. Our novel *BACE1* and *SIDT2* findings were confirmed using summary statistics from the Global Lipids Genetic Consortium (GLGC) meta-GWAS.

# INTRODUCTION

Though 99.9% of the genome between humans is identical, millions of variant sites exist in different frequencies between populations, which leads to differences in gene expression and other complex traits (*Brown et al., 2016*). Since most GWAS have been conducted in European cohorts and most databases are built upon European data, the results may not accurately extrapolate to other global populations, which could lead to disparity within medicine (*Bustamante, De La Vega & Burchard, 2011*). This discrepancy is alarming

Corresponding author
Heather E. Wheeler,
hwheeler1@luc.edu

considering urgent health issues worldwide, such as obesity and cardiovascular disease. Lipid levels as a complex trait are increasingly concerning due to the growing global rate of obesity caused by the increasing availability of high-fat foods and rapid urbanization (*Ellulu et al., 2014*). Decreased high density lipoprotein (HDL) cholesterol levels and increased low density lipoprotein (LDL) cholesterol and triglyceride (TRIG) levels are associated with cardiovascular disease, the leading cause of death in the United States (*Go et al., 2013*). Studies such as the Global Lipids Genetic Consortium (GLGC) acquire information predominantly from Europeans, but lack information from other populations (*Coram et al., 2015*; *Willer et al., 2013*). We aim to help remedy this issue by studying lipid traits in diverse populations. We chose two populations with lipid phenotypes available to study from the database of Genotypes and Phenotypes: Yoruba in Ibadan, Nigeria, (Yoruba) and Filipino in Cebu, Philippines (Cebu) (*Hall et al., 2006*; *Adair et al., 2011*; *Wu et al., 2013*).

At the time of our study, one of the largest available cholesterol SNP meta-analyses is the GLGC (*Willer et al., 2013*). The cohort in that study consists of 188,577 European-ancestry individuals and 7,898 non-European-ancestry individuals. One hundred fifty-seven loci were found to be significantly associated with total cholesterol (CHOL), HDL, TRIG, or LDL levels, and they conducted further gene set enrichment analysis with MAGENTA (*Ayellet et al., 2010*). However, gene-level association studies that integrate transcriptome data, like PrediXcan and TWAS, were not performed (*Gamazon et al., 2015*; *Gusev et al., 2016*). Summary statistics from GLGC were used as a replication and base set in our analyses of the Yoruba and Cebu cohorts.

Both the Cebu and Yoruba cohorts have been used in genetic studies of lipids previously (*Hall et al., 2006*; *Wu et al., 2013*). These studies, both using the same data we study here, focused on *APOE*, a well-known gene that is associated with lipid levels and Alzheimer's disease (*Middelberg et al., 2011*). Previously, SNP-level GWAS in the Cebu and other East Asian cohorts attributed rs662799 as affecting function of *APOA5*, which is located 571 bases upstream (*Wu et al., 2013*; *Lu et al., 2016*; *Spracklen et al., 2017*). Beyond the *APOE* candidate gene study, no full GWAS has been conducted in the Yoruba cohort.

In this study, we performed a genome wide association study (GWAS) in each population using linear mixed modeling (*Zhou & Stephens, 2012*) and a conditional and joint analysis (*Yang et al., 2012*). Subsequently, we calculated the genetic correlation for each lipid trait between the populations at the SNP-level using bivariate REML analysis (*Yang et al., 2011*). We also used cross-population empirical Bayes (XPEB) modeling to improve power to detect SNPs with similar effects as previously found in larger European meta-analyses (*Coram et al., 2015*). Finally, we used the transcriptome-informed method PrediXcan to implicate genes in CHOL, HDL, LDL, and TRIG (*Gamazon et al., 2015*). Using our data and those from previous European studies, we confirm previously known loci and implicate new loci through the mechanism of gene expression regulation in Filipinos (*Willer et al., 2013*). Our gene-based association study for triglyceride levels in the Cebu cohort suggests that rs662799 may affect the expression of *BACE1* and *SIDT2* rather than that of *APOA5*.
**Table 1 Data analyzed.**

|  | Yoruba | Cebu |
|---|---|---|
| Accession number | phs000378.v1.p1 | phs000523.v1.p1 |
| Type of genotyping | Whole genome genotyping | Whole genome genotyping |
| Source platform | Illumina, HumanOmni2.5 | Affymetrix, Genomewide Human SNP Array 5.0 |
| Pre-QC SNPs | 2,443,179 | 440,792 |
| Pre-QC individuals | 1,251 | 1,799 |
| Post-QC SNPs | 1,522,836 | 369,185 |
| Post-QC individuals | 1,017 | 1,765 |
| Post-imputation GWAS SNPs | 12,553,142 | 4,496,603 |

## MATERIALS AND METHODS

### Cohorts

We obtained data from both cohorts through the database of Genotypes and Phenotypes with Institutional Review Board approval (*Mailman et al., 2007*) (Table 1). Yoruba consists of 1,251 adults of Yoruba ethnicity age 73 to 103 years old, living in Ibadan, Nigeria in 2001, who were originally studied in the Indianapolis Ibadan Epidemiological Study of Dementia (*Ogunniyi et al., 1997*). The Cebu population consists of 1,799 Filipino mothers, who gave birth between May 1, 1983 and April 30, 1984 in the metropolitan area of Cebu, Philippines. This cohort was originally studied in the Cebu Longitudinal Health and Nutrition Survey (*Adair et al., 2011*), and at the time of data collection in 2005, the mothers were age 34 to 70. The Yoruba cohort was genotyped with the Illumina HumanOmni2.5 array and the Cebu cohort was genotyped with the Affymetrix Genomewide Human SNP Array 5.0. Both cohorts had CHOL, HDL, LDL, and TRIG levels measured after fasting (mg/dL) and we subsequently rank normalized each trait (*Aulchenko et al., 2007*) (Table S2). See https://github.com/aandaleon/px_chol for all scripts used in our analyses.

Summary statistics from the Global Lipids Genetic Consortium meta-analysis (*Willer et al., 2013*) were downloaded from http://csg.sph.umich.edu/abecasis/public/lipids2013/. Though the offspring of the Cebu cohort are included in the GLGC cohort, they form a small portion of the dataset (1,771/188,577) and thus do not drive the signal for the entire dataset.

### Quality control

We performed quality control on the genotypes in these cohorts with PLINK following a standard quality control pipeline (*Turner et al., 2001*; *Purcell et al., 2007*). Starting with the dbGaP PLINK binary files, we removed SNPs with call rates < 99% in the individual populations. Subsequently, SMARTPCA within Eigensoft was used to map individuals on their first 10 principal components, and individuals with excess (±5 standard deviations) from the population mean on the first two components were removed (*Patterson, Price & Reich, 2006*) (Figs. S1 and S2). This was followed by removing individuals with excess heterozygosity (±3 standard deviations), leaving the Yoruba cohort with 1,017 individuals with genotype and lipid phenotype data, including 1,522,836 SNPs. The Cebu cohort

retained 1,757 individuals with genotype and lipid phenotype data, including 369,185 SNPs. Both cohorts were mapped to hg19, which included performing a liftover in the Cebu cohort from hg18 to hg19. We used an imputation preparation tool on the Cebu cohort available at http://www.well.ox.ac.uk/~wrayner/tools/ that adjusted the data by matching the strand, alleles, position, ref/alt assignments, and frequency differences to the 1000G reference panel.

## Imputation

Yoruba SNPs were imputed on the Sanger Imputation Server with EAGLE2 and PBWT using the African Genomes Reference Panel to improve genome coverage (*Delaneau, Marchini & Zagury, 2012*; *Durbin, 2014*; *McCarthy et al., 2016*). We imputed the Cebu SNPs using the Michigan Imputation Server with the 1000 Genomes phase 3 reference panel and EAGLE2 (*Auton et al., 2015*; *Das et al., 2016*; *Loh et al., 2016*). The output from the imputation was filtered to remove SNPs with $R^2 < 0.8$ and minor allele frequency <0.01, leaving 12,553,142 SNPs for analysis in Yoruba and 4,496,603 SNPs for analysis in Cebu.

## SNP-level genome-wide association study

The imputed genotype dosages were used in a genome-wide association study performed with Genome-Wide Efficient Mixed Model Analysis (GEMMA) software using a univariate linear mixed model for each of the four phenotypes (*Zhou & Stephens, 2012*). SNPs with $P < 5 \times 10^{-8}$ using the Wald test were considered genome-wide significant. The top SNPs from the GEMMA analysis were plotted using LocusZoom to depict their proximity to various genes (*Pruim et al., 2011*). For each phenotype tested, we also used GEMMA to obtain the percent variance explained (PVE) by all the SNPs, i.e., the "chip heritability" (*Zhou & Stephens, 2012*). Conditional and joint analyses were then performed using GCTA-COJO (*Yang et al., 2011*; *Yang et al., 2012*) to identify the lead SNP or SNPs at each locus.

## Comparison of populations

Both populations' GEMMA results and summary statistics from GLGC (*Willer et al., 2013*) were used in a cross-population empirical Bayes model (XPEB) to compute false discovery rates for each SNP, with significance declared at FDR < 0.05. This model improves efficiency in GWAS by incorporating relevant results from larger (ex. GLGC) GWAS only when there are similar effect sizes between populations (*Coram et al., 2015*). Sample sizes in the GLGC results ranged from 50,000 to 187,365 depending on the SNP. Because XPEB assumes similar sample size in the base population across SNPs, we ran XPEB using only the SNPs with sample size between 80,000 and 95,000 in GLGC, which left us with 4,454,201 markers for each phenotype. SNP-level comparisons between populations were performed using Genome-wide Complex Trait Analysis (GCTA) software (*Yang et al., 2011*). We performed a bivariate restricted maximum likelihood (REML) analysis to estimate the genetic correlation between the Cebu and Yoruba cohorts for each phenotype (*Lee et al., 2012*).

**Table 2** **The number of genes tested in PrediXcan using expression prediction models built in GTEx Project tissues.** Genes tested had a cross-validated prediction performance $R^2 > 0.01$.

| Tissue model | Tissue abbreviation | Genes tested | Tissue model | Tissue abbreviation | Genes tested |
|---|---|---|---|---|---|
| Adipose—Subcutaneous | ADPSBQ | 7,254 | Esophagus—Mucosa | ESPMCS | 7,710 |
| Adipose—Visceral (Omentum) | ADPVSC | 4,447 | Esophagus—Muscularis | ESPMSL | 6,338 |
| Adrenal Gland | ADRNLG | 3,785 | Heart—Atrial Appendage | HRTAA | 4,450 |
| Artery—Aorta | ARTAORT | 5,943 | Heart—Left Ventricle | HRTLV | 4,718 |
| Artery—Coronary | ARTCRN | 3,141 | Liver | LIVER | 2,502 |
| Artery—Tibial | ARTTBL | 7,074 | Lung | LUNG | 6,448 |
| Brain—Anterior cingulate cortex (BA24) | BRNACC | 2,430 | Muscle—Skeletal | MSCLSK | 6,520 |
| Brain—Caudate (basal ganglia) | BRNCDT | 3,325 | Nerve—Tibial | NERVET | 8,016 |
| Brain—Cerebellar hemisphere | BRNCHB | 4,077 | Ovary | OVARY | 2,673 |
| Brain—Cerebellum | BRNCHA | 5,066 | Pancreas | PNCREAS | 4,603 |
| Brain—Cortex | BRNCTXA | 3,334 | Pituitary | PTTARY | 3,094 |
| Brain—Frontal Cortex (BA9) | BRNCTXB | 3,138 | Prostate | PRSTTE | 2,491 |
| Brain—Hippocampus | BRNHPP | 2,508 | Skin—Not Sun Exposed (Suprapubic) | SKINNS | 5,471 |
| Brain—Hypothalamus | BRNHPT | 2,290 | Skin—Sun Exposed (Lower leg) | SKINS | 7,665 |
| Brain—Nucleus accumbens (basal ganglia) | BRNNCC | 2,984 | Small Intestine—Terminal Ileum | SNTTRM | 2,515 |
| Brain—Putamen (basal ganglia) | BRNPTM | 2,621 | Spleen | SPLEEN | 3,602 |
| Breast—Mammary Tissue | BREAST | 4,473 | Stomach | STMACH | 4,035 |
| Cells—EBV-transformed lymphocytes | LCL | 3,441 | Testis | TESTIS | 7,002 |
| Cells—Transformed fibroblasts | FIBRBLS | 7,543 | Thyroid | THYROID | 7,853 |
| Colon—Sigmoid | CLNSGM | 3,619 | Uterus | UTERUS | 2,058 |
| Colon—Transverse | CLNTRN | 4,729 | Vagina | VAGINA | 1,939 |
| Esophagus—Gastroesophageal Junction | ESPGEJ | 3,457 | Whole Blood | WHLBLD | 6,588 |

## Gene-based association study

PrediXcan, the gene-level association study, was performed using models built with cis-expression quantitative trait loci results from the Genotype-Tissue Expression Project (GTEx) (*Ardlie et al., 2015*; *Wheeler et al., 2016*; *Barbeira et al., 2017*). GTEx models (GTEx-V6p-HapMap-2016-09-08.tar.gz) were downloaded from PredictDB at http://predictdb.hakyimlab.org/. Significance for each tissue was determined as FDR < 0.1 across all testable genes in all tissues ($N = 198,970$). In total, 44 GTEx models were tested in both cohorts (Table 2). Predicted expression levels were obtained and tested for association with the lipid phenotypes using PrediXcan software (*Gamazon et al., 2015*). Significant genes were further plotted using ggplot to depict the predicted gene expression against the observed phenotype (*Wickham, Winston & RStudio, 2016*). For the replication cohort, GLGC, Summary-PrediXcan (*Barbeira et al., 2017*) was used because only summary statistics were available.

## Backward elimination modeling

Because our PrediXcan analysis showed multiple genes associated with TRIG at the 11q23.3 locus in Cebu, we conducted a backward elimination analysis to determine the lead gene or genes. We used the R *lm* function to build all multiple linear regression models. The

**Table 3  SNP-level GWAS results in the Yoruba (Y) and Cebu (C) populations.** Shown are SNPs that reach genome-wide significance after conditional and joint analysis (*Yang et al., 2011*; *Yang et al., 2012*).

| Pop. | Pheno. | Chr. | Position | SNP ids | Nearest gene | Non-effect allele | Effect allele | EAF | Marginal Beta | Marginal P | Joint Beta | Joint P |
|------|--------|------|----------|---------|--------------|-------------------|---------------|------|---------------|------------|------------|---------|
| Y | HDL | 16 | 56995935 | rs34065661 | *CETP* | C | G | 0.093 | 0.468 | $9.0 \times 10^{-10}$ | 0.468 | $8.7 \times 10^{-10}$ |
| Y | LDL | 19 | 45413233 | rs1065853 | *APOE* | G | T | 0.113 | −0.405 | $6.6 \times 10^{-9}$ | −0.405 | $6.4 \times 10^{-9}$ |
| C | TRIG | 11 | 116663707 | rs662799 | *APOA5* | A | G | 0.245 | 0.342 | $2.7 \times 10^{-16}$ | 0.342 | $2.5 \times 10^{-18}$ |
| C | TRIG | 2 | 27731212 | rs3817588 | *GCKR* | T | C | 0.309 | −0.202 | $2.6 \times 10^{-7}$ | −0.202 | $2.1 \times 10^{-8}$ |

starting model included predicted expression terms for all genes with rs662799 or a linked SNP ($r^2 > 0.6$) in its predictive model and the absolute value of the marginal $t$-statistic greater than 3. The term with the highest $P$-value was eliminated and the model rerun until only terms with $P < 0.05$ remained in the model.

# RESULTS

## Yoruba SNP-level GWAS

We sought to better understand the genetic architecture of lipid traits within and across populations. In the Yoruba cohort from Ibadan, Nigeria, which included 1,017 individuals and 12,553,142 SNPs, we conducted SNP-level GWAS for four lipid traits CHOL, HDL, LDL, and TRIG. For each lipid trait, we used a univariate linear mixed model, which accounts for relatedness within the populations (*Zhou & Stephens, 2012*). This was especially important because one-third of the Yoruba cohort is related to at least one other member (proportion identity by descent $> 0.125$). Across the four phenotypes, five SNPs surpassed the genome-wide significance threshold of $P < 5 \times 10^{-8}$ at two loci (Fig. 1). Conditional and joint analysis (*Yang et al., 2011*; *Yang et al., 2012*) did not reveal additional associated SNPs at these two loci and the top hits are shown in Table 3. rs34065661 is on chromosome 16q13 within an intron of *CETP* and rs1065853 is on chromosome 19q13.32 near *APOE* (Fig. 1). Both *CETP* and *APOE* are well-known and well-studied lipid genes (*Buyske et al., 2012*; *Rasmussen-Torvik et al., 2012*).

## Cebu SNP-level GWAS

We performed SNP-level GWAS for the same four lipid phenotypes in 1,765 individuals from Cebu, Philippines, using 4.5 million imputed SNPs (see Methods). No SNPs met genome-wide significance for the CHOL, HDL, and LDL phenotypes. However, 44 SNPs were genome-wide significant ($P < 5 \times 10^{-8}$) for TRIG and all grouped on chromosome 11q23.3 (Fig. 2), a locus that includes various lipid genes such as *APOA1* and *APOA4*. The most significant SNP in this group is rs662799, with a marginal $P = 2.7 \times 10^{-16}$. It is located 571 base pairs upstream of *APOA5*, and has been previously associated with cholesterol traits in Asian populations, possibly due to its high minor allele frequency within Asian populations, with MAF for minor allele G at 0.245 in 1000 Genomes EAS compared to 0.083 in EUR (*Marcus & Novembre, 2016*) (Fig. 3). Conditional and joint analysis (*Yang et al., 2011*; *Yang et al., 2012*) did not reveal additional associated SNPs at the 11q23.3 locus,

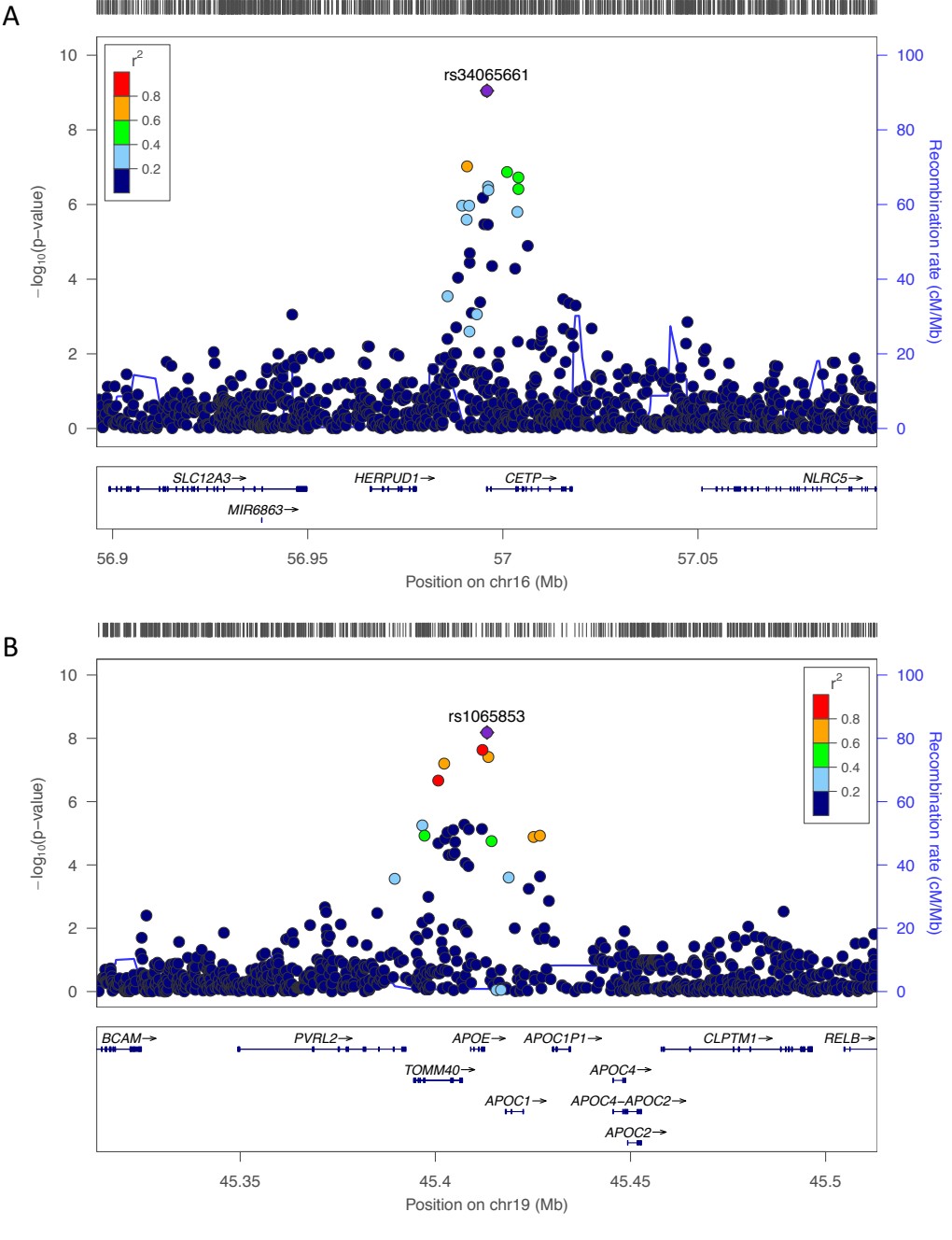

**Figure 1  LocusZoom plots of the most significant SNPs in (A) HDL (rs34065661) and (B) LDL (rs1065853) in Yoruba.** The color of each dot represents the SNP's linkage disequilibrium $r^2$ with the labeled SNP in the 1000 Genomes African populations.

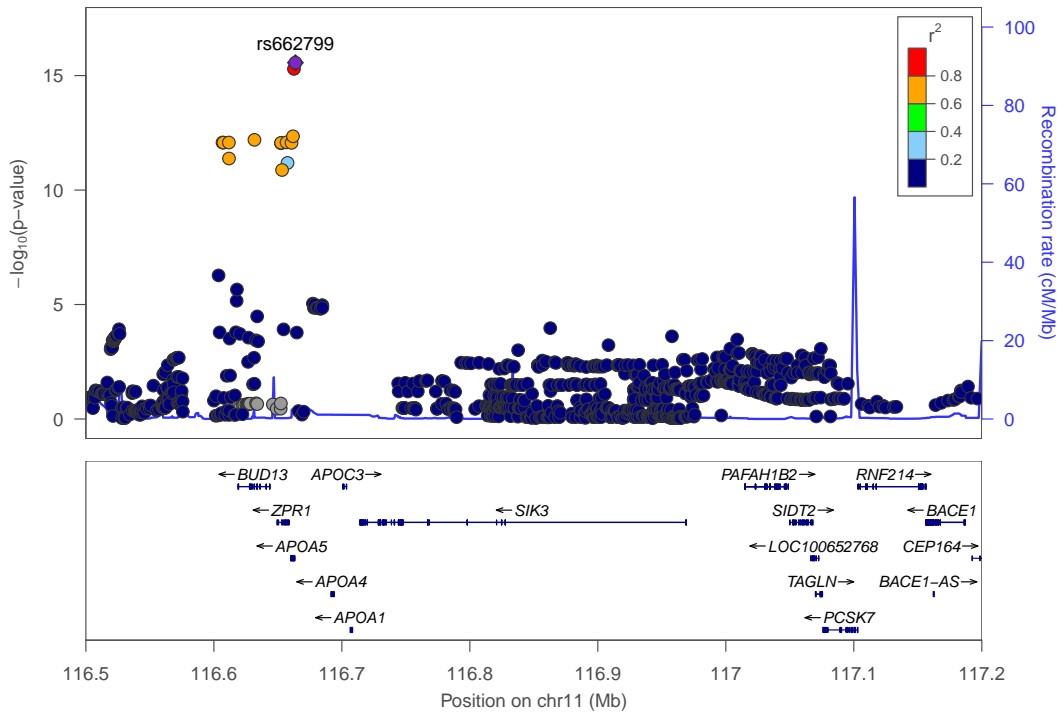

**Figure 2** **The top Cebu GWAS signal, rs662799, which associated with TRIG levels is 571 bp upstream of *APOA5*.** The color of each dot represents the SNP's linkage disequilibrium $r^2$ with rs662799 in the 1000 Genomes East Asian populations.

but did reveal an additional genome-wide significant SNP on chromosome 2 in the *GCKR* gene (Table 3). As a positive control, we compared the results from the previous Cebu GWAS (*Wu et al., 2013*) and our GWAS and obtained largely the same significant results (Table S1).

## Integrating larger European study results into Yoruba and Cebu SNP-level GWAS

The overlapping genetic architecture between populations for most traits is likely nonzero, but not 100% either due to differences in allele frequencies, effect sizes, and linkage disequilibrium patterns. Currently, European GWAS often have sample sizes 100 times larger than non-European studies. However, studies in diverse populations are growing (*Spracklen et al., 2017*; *Wojcik et al., 2017*). Traditional meta-analysis methods give the most weight to the GWAS with the largest sample size. Therefore, meta-analyses combining populations by traditional methods would be driven by the European results, drowning out any additional signal. The cross-population empirical Bayes (XPEB) method is designed to boost signal in a target (small, usually non-European) population whenever the base (large, usually European) population shares genetic architecture, but does not generate false positives when the signal is only present in the base population (*Coram et al., 2015*).

We used XPEB to improve power for mapping lipid traits in the Yoruba and Cebu cohorts by integrating results from the base population GLGC, a large lipid meta-analysis

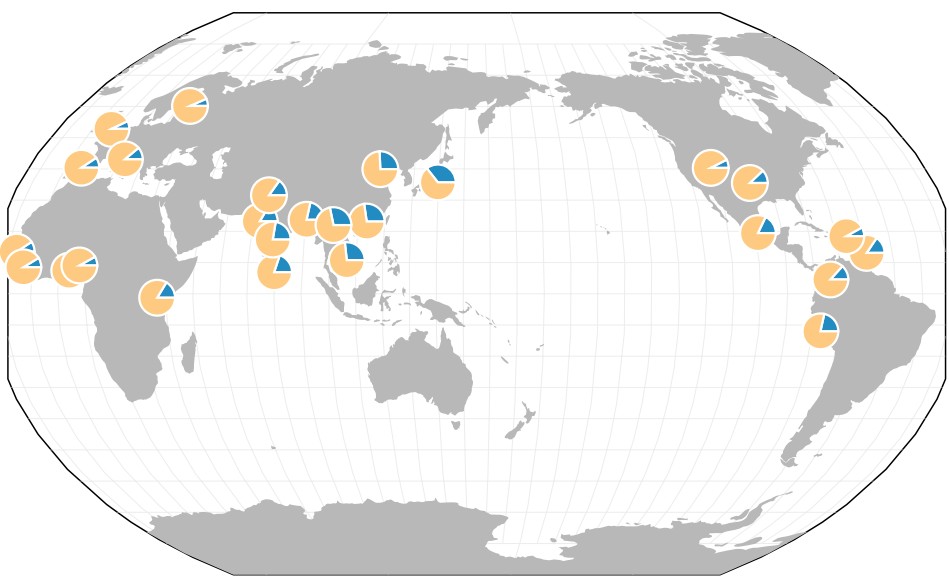

chr11:116663707 G/A

**Figure 3** **Allele frequencies of TRIG associated SNP and driver of predicted expression models in multiple genes, rs662799, in 1000 Genome populations.** Figure generated with the Geography of Genetic Variants Browser (*Marcus & Novembre, 2016*).

of European individuals (*Willer et al., 2013*). In XPEB, we input the SNPs and *P*-values from both our target GWAS (Yoruba or Cebu) and the base European GWAS. The output includes a population-wide estimate of the degree of genetic architecture overlap, $\kappa_1$, and a new false discovery rate (FDR) for each individual SNP in common between the base and target GWAS.

In the Yoruba population, we found associated loci (FDR < 0.05) using XPEB for the CHOL, HDL, and LDL phenotypes (Table S3). These results reflect the estimated architecture overlap with GLGC, where $\kappa_1$ was 0.65 for CHOL, 0.51 for HDL, 0.9 for LDL, and 0 for TRIG. The *CETP* locus, which was also significant in the Yoruba-only HDL GWAS was the most significant result in the XPEB analysis. In addition, several other previously implicated genes were significant in the XPEB analysis, including *LDLR* for CHOL and *PCSK9*, *LPA*, and *SMARCA4* for LDL (*Wu et al., 2013*; *Willer et al., 2013*; *Surakka et al., 2014*).

In Cebu, CHOL, TRIG, and LDL each had $\kappa_1 = 0.90$, while HDL had $\kappa 1 = 0.64$. Tens or hundreds of SNPs had FDR < 0.05 for each phenotype, including those found in the Cebu-only GWAS (Table 3). Additional significant SNPs located within or near other well-known, previously studied lipid genes included *CETP* and *LIPC* in HDL; *PLCG1* and *TOP1* in LDL; and *APOA5* and *BUD13* in TRIG (*Asselbergs et al., 2012*; *Spracklen et al., 2017*; *Wu et al., 2013*; *Zhou et al., 2013*; *Kim et al., 2011*) (Table S4).

**Table 4** Percent variance explained (PVE) and standard error as estimated in GEMMA for each trait compared.

| | Yoruba ($n = 1,017$) | Cebu ($n = 1,765$) | European ($n = 5,123$)[a] |
|---|---|---|---|
| CHOL | $0.040 \pm 0.061$ | $0.067 \pm 0.093$ | $0.29 \pm 0.043$ |
| HDL | $0.013 \pm 0.037$ | $0.217 \pm 0.091$ | $0.34 \pm 0.043$ |
| TRIG | $0.049 \pm 0.110$ | $0.140 \pm 0.103$ | $0.38 \pm 0.041$ |
| LDL | $0.029 \pm 0.046$ | $0 \pm 0.096$ | $0.19 \pm 0.047$ |

**Notes.**

[a] *Sabatti et al. (2009)* and *Zhou (2017)*.

## Heritability

As part of our GWAS study, we also estimated the percent variance explained (PVE) by all SNPs tested, i.e., "chip heritability", using GEMMA and conducted further genetic correlation analysis using GCTA (*Yang et al., 2011*; *Zhou & Stephens, 2012*). By estimating heritability, we can help determine which portion of our phenotype is not explained through our analyses and is influenced by other factors, such as diet.

In the Yoruba cohort, no PVE estimate was significantly different than zero. All PVE estimates for Yoruba were low when compared to variance component studies in a Finnish cohort of 5,123 individuals (*Sabatti et al., 2009*; *Zhou, 2017*). Unlike in Yoruba, PVE estimates for two phenotypes, HDL and TRIG, within the Cebu were significantly different than zero and closer to the Finnish estimates (Table 4).

We attempted to estimate the genetic correlation between all SNPs in the Cebu and Yoruba populations using bivariate REML analysis as implemented in GCTA (*Lee et al., 2012*). The only phenotype that converged was TRIG, with an estimated correlation of $0.644 \pm 0.65$, indicating there is shared architecture between the populations. However, the small sample sizes available in our study do not offer enough power to reliably estimate heritability and genetic correlation as indicated by the large standard errors.

## Yoruba gene-based association study

While many GWAS have been performed on lipid traits, most of the significant SNPs found fall outside of protein coding regions and thus their mechanisms of action are not immediately apparent. PrediXcan is a gene-level association method that incorporates functional data on potential regulatory elements to provide mechanistic directionality for association of a gene with a phenotype (*Gamazon et al., 2015*). PrediXcan uses gene expression prediction models built from genome-transcriptome datasets such as the Genotype-Tissue Expression (GTEx) Project to predict gene expression from genotype and then tests the predicted expression levels for association with with trait of interest (*Ardlie et al., 2015*). We applied PrediXcan to our SNP-level GWAS results using models built in 44 GTEx tissues (*Barbeira et al., 2017*) (Table 2).

For each tissue, we declared associations significant if FDR $< 0.1$, across all genes and tissues tested, to adjust for multiple testing. Of the four phenotypes for this cohort and 44 tissues with models available, one gene, *PAX6*, surpassed the significance threshold. *PAX6* was not significant in Cebu or GLGC (Table 5). In humans, *PAX6* has been associated with

**Table 5   Top genes (FDR < 0.1) in Yoruba and Cebu found using PrediXcan.** All results are for the TRIG phenotype. Results from both populations and the Global Lipids Genetic Consortium (GLGC) are shown. Six associations discovered in Cebu replicated in the GLGC.

| Chr. | Tissue | Gene name | Beta (Yoruba) | P (Yoruba) | FDR (Yoruba) | Beta (Cebu) | P (Cebu) | FDR (Cebu) | Beta (GLGC) | P (GLGC) |
|---|---|---|---|---|---|---|---|---|---|---|
| 2p23.3 | THYROID | *FNDC4* | 0.05 | 0.50 | 0.99 | −0.24 | $1.0 \times 10^{-6}$ | 0.037 | −0.11 | $7.7 \times 10^{-83}$ |
| 11p13 | ARTAORT | *PAX6* | −0.74 | $4.8 \times 10^{-7}$ | 0.093 | 0.16 | 0.20 | 0.98 | 0 | 0.91 |
| 11q23.3 | ESPMCS | *BACE1* | 0.22 | 0.33 | 0.99 | −3.30 | $1.7 \times 10^{-15}$ | $3.0 \times 10^{-10}$ | −0.17 | $7.3 \times 10^{-19}$ |
| 11q23.3 | CLNSGM | *APOA4* | 0.80 | 0.55 | 0.99 | −11.6 | $6.5 \times 10^{-12}$ | $5.9 \times 10^{-7}$ | −1.95 | $1.7 \times 10^{-39}$ |
| 11q23.3 | BRNHPP | *APOA1* | 0.015 | 0.88 | 0.99 | 0.50 | $3.2 \times 10^{-7}$ | 0.019 | 0 | 0.31 |
| 11q23.3 | HRTLV | *SIDT2* | 0.24 | 0.21 | 0.98 | 0.90 | $8.1 \times 10^{-7}$ | 0.037 | 0.16 | $9.3 \times 10^{-29}$ |
| 11q23.3 | THYROID | *SIDT2* | 0.10 | 0.61 | 0.99 | −0.55 | $1.7 \times 10^{-6}$ | 0.050 | −0.55 | $1.1 \times 10^{-102}$ |
| 11q23.3 | WHLBLD | *BACE1* | 0.00 | 0.99 | 0.99 | −0.79 | $3.4 \times 10^{-6}$ | 0.087 | −0.28 | $4.0 \times 10^{-28}$ |

insulin production (*Ahlqvist et al., 2012*). Currently, there is little known about *PAX6* and its potential association with lipid or cardiovascular phenotypes.

## Cebu gene-based association study

When we applied PrediXcan to the Cebu cohort, seven genes were found significant (Table 5). A few of these genes, such as *FNDC4*, *APOA1* and *APOA4*, are well-documented in lipid traits, while genes such as *SIDT2* have been previously implicated in Asians (*Teslovich et al., 2010*; *Kim et al., 2011*; *Willer et al., 2013*; *Wu et al., 2013*; *Zhou et al., 2013*; *Gombojav et al., 2015*; *Lu et al., 2016*; *Spracklen et al., 2017*). The association of *BACE1* with TRIG was highly significant (FDR $= 3.02 \times 10^{-10}$, Fig. 4). The predicted increase in *BACE1* expression and decrease in TRIG levels is an association not previously seen in humans, but has been observed in mice (*Meakin et al., 2012*; *Baek et al., 2016*) (Fig. 5). Additionally, *SIDT2* has a similar effect across many tissue models, indicating its potential importance in regulating TRIG levels as well (Figs. 5 and 6). Both *BACE1* and *SIDT2* have increased gene expression associated with decreased TRIG levels in most tissues and share many SNPs in most of their prediction models.

We conducted a backward elimination analysis to determine the lead gene or genes at the 11q23.3 locus. The starting model include all gene-tissue combinations in Fig. 6, which includes all genes with rs662799 or a linked SNP ($r^2 > 0.6$) in its predictive model and $|t| > 3$, where $t$ is the association test statistic. The most significant gene in the final model was ESPMCS-*BACE1* ($P = 1 \times 10^{-10}$), with residual effects present ($P < 0.05$) for LCL-*SIDT2*, MSCLSK-*APOA1*, TESTIS-*CEP164*, and WHLBLD-*BACE1* (Table 6).

## Comparing populations

In a direct comparison of the significant genes obtained through PrediXcan for the Cebu and Yoruba cohorts, there is no gene that shares significance (FDR < 0.10) between populations (Table 5). There was no overlap in significant genes between Yoruba and GLGC, but there was overlap in significant genes between Cebu and GLGC in *FNDC4*, *SIDT2*, *APOA4*, and *BACE1* with the same effect direction for all genes (*Willer et al., 2013*) (Table 5).

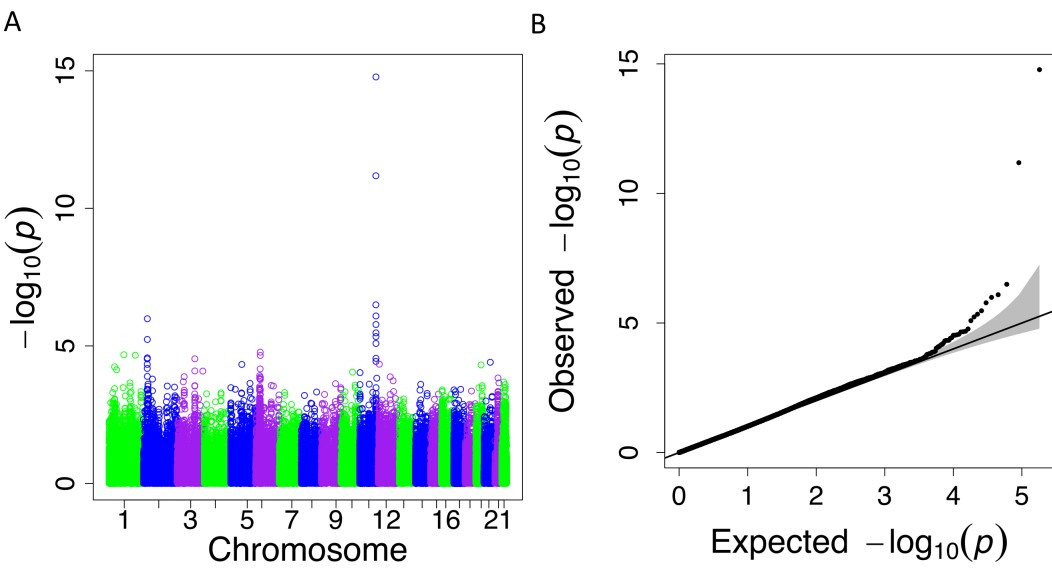

**Figure 4** **PrediXcan results for the Cebu TRIG phenotype using gene expression models built in 44 GTEx tissues.** (A) Manhattan plot: $-\log_{10}$ $P$-values are plotted against the respective chromosomal position of each gene across all tissues. (B) QQ plot of observed versus expected $-\log_{10}$ $P$-values for each gene across all tissues.

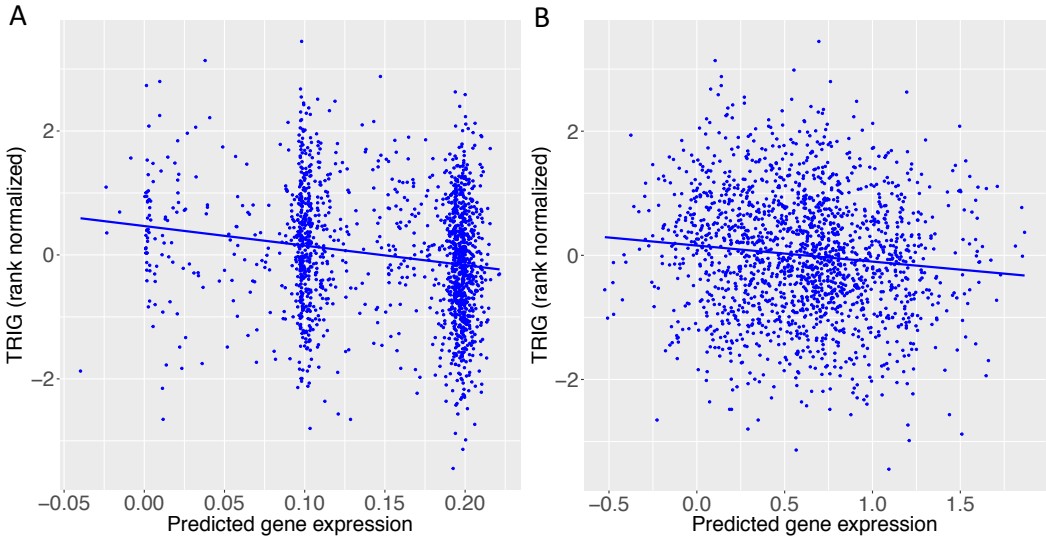

**Figure 5** **TRIG levels vs. predicted expression of two genes in Cebu.** (A) *BACE1* predicted expression using the GTEx ESPMCS prediction model. (B) *SIDT2* predicted expression using the GTEx LCL prediction model.

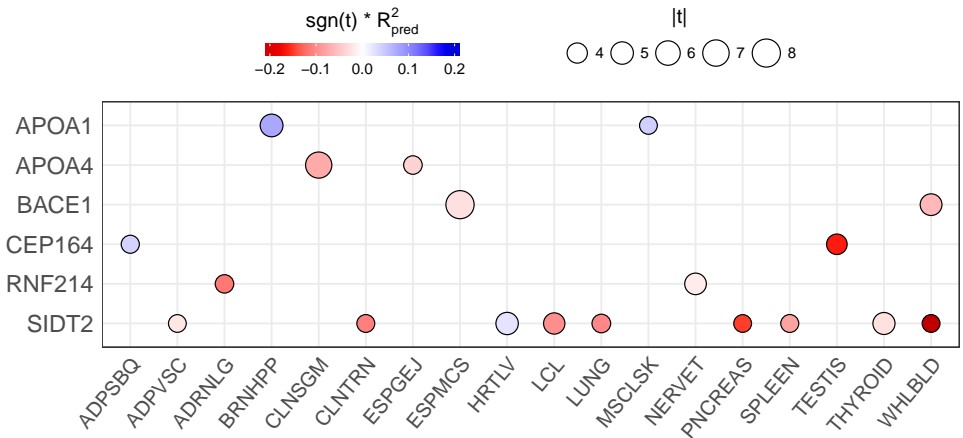

**Figure 6** **PrediXcan results of genes with rs662799 or linked SNPs ($r^2 > 0.6$) in their predictive models across tissues.** The size of the circle is proportional to the absolute value of the $t$-statistic in the PrediXcan association test and the color indicates the direction of effect (sign of the t-statistic, $-1$ or $1$) multiplied by the prediction performance of the model ($R^2$) for each gene-tissue combination. Only genes $|t| > 3$ are plotted for clarity.

**Table 6** **Gene-tissue combinations from Fig. 6 with $P < 0.05$ in a backwards-elimination linear model including the TRIG phenotype and predicted gene expression terms.**

| Tissue | Gene | Estimate | Std. Error | $t$ value | $P$ |
|--------|------|----------|-----------|-----------|-----|
| ESPMCS | *BACE1* | −3.7 | 0.57 | −6.4 | $1.7 \times 10^{-10}$ |
| LCL | *SIDT2* | −0.20 | 0.06 | −3.2 | 0.001 |
| MSCLSK | *APOA1* | 1.5 | 0.52 | 2.9 | 0.003 |
| TESTIS | *CEP164* | −0.17 | 0.08 | −2.2 | 0.025 |
| WHLBLD | *BACE1* | 0.83 | 0.29 | 2.9 | 0.003 |

## DISCUSSION

Using genome-wide genotypes and lipid levels obtained in two diverse populations from Ibadan, Nigeria and Cebu, Philippines, we performed multiple genome-wide analyses with the goal of better understanding the underlying genetic architecture of cholesterol traits in both populations.

### Top GWAS SNPs have been previously shown to associate with lipid traits

Within the Yoruba portion of our study, rs34065661 and rs17231520, SNPs close to or within *CETP*, were significant (Fig. 1A). *CETP*, cholesteryl ester transfer protein, is a well-known lipid gene previously implicated in lipid studies in African populations (*Buyske et al., 2012*; *Elbers et al., 2012*). *CETP* is involved in the transfer of cholesterol from HDL to other lipoproteins, and is a common genetic target for statins and other cholesterol-lowering drugs (*Barter et al., 2003*). It also has strong association with Alzheimer's disease and other neurodegenerative diseases (*Xiao et al., 2012*). Additional significant SNPs in Yoruba, rs1065853, rs7412, and rs75627662, are located near or within *APOE* (Fig. 1B). *APOE*,

apolipoprotein E, is another well-known cholesterol gene previously implicated in other lipid studies (*Rasmussen-Torvik et al., 2012*; *Surakka et al., 2014*; *Mahley, 2016*; *Spracklen et al., 2017*; *Zhu et al., 2017*). It acts as a lipid transport protein in high association with LDL receptors and is also strongly associated with neurodegenerative diseases (*Moriarty et al., 2017*).

The most significant SNP within the Cebu GWAS portion of our study is rs662799, at $P = 2.7 \times 10^{-16}$. It has been previously associated with cholesterol traits in Asian populations (*Lu et al., 2016*; *Spracklen et al., 2017*) (Fig. 2). *APOA5* is a well-documented gene associated with triglyceride levels (*Go et al., 2013*). Other genes within this linkage group include *ZPR1* and *BUD13*, which have both been previously implicated with triglyceride levels in East Asians as well, possibly due to its higher minor allele frequency in those populations (*Kim et al., 2011*; *Lin et al., 2016*) (Fig. 3).

## The mechanism underlying the association of rs662799 with TRIG levels may include long distance regulation of *BACE1* and *SIDT2*

In our PrediXcan analysis, *BACE1* reached significance (FDR $= 3.0 \times 10^{-10}$) in both the Cebu and GLGC replication cohorts, but it was not significant in Yoruba (Fig. 4). Currently, in humans, *BACE1* is known to increase risk of Alzheimer's disease with increased expression (*Cole & Vassar, 2007*). While not recognized as significant in the GLGC SNP meta-analysis (*Willer et al., 2013*), our application of PrediXcan to the GLGC GWAS results verified that the *BACE1* association with TRIG is significant ($P = 7.3 \times 10^{-19}$) (Table 5).

This gene has been studied in terms of Alzheimer's disease and weight gain in mice. In our results, we found increased predicted gene expression for *BACE1* to be associated with lower TRIG levels (Table 5). For *BACE1* knockout mice, there was no significant difference in triglyceride levels versus wild-type mice, but they did have lower average body weight (*Meakin et al., 2012*). Additionally, in mice, higher triglyceride levels were found to reduce *BACE1* expression in a study concerning Alzheimer's treatment (*Baek et al., 2016*). This latter result is consistent with our finding that increased *BACE1* expression is associated with lower triglyceride levels (Fig. 5).

The SNP rs662799 is 571 bases upstream of the gene with which it is typically associated, *APOA5* (Fig. 2). *APOA5* was well-predicted in only one tissue, SNTTRM, and the prediction was not driven by SNPs linked to rs662799. Thus, *APOA5* may not affect TRIG levels through the mechanism of variation in gene expression regulation. Even though rs662799 is located 493kb downstream of *BACE1*, it has the largest effect size, i.e., it is the driver SNP, in the predictive model for *BACE1* in ESPMCS. SNPs closer to *BACE1* are not linked to rs662799 (Fig. 2). The similar effects and significance of *BACE1* in mouse studies, the Cebu cohort analysis, and GLGC PrediXcan results indicate the increased expression of *BACE1* is associated with decreased TRIG levels and that variation in the regulation of *BACE1* may contribute to differences in TRIG levels.

Another significant gene in our gene-based association study of TRIG in Cebu is *SIDT2*, which is 386 kb downstream of rs662799. Here, the effects of many SNPs, including some linked to rs662799, combine in the prediction model for *SIDT2* gene expression, with

no dominant driver SNP, as demonstrated by the lack of discernible clusters in the plot of TRIG levels versus predicted expression (Fig. 5). Additionally, *SIDT2* exhibits more consistent effect sizes over more tissues than *BACE1* (Fig. 6), in which predicted expression is associated with higher TRIG levels (Fig. 5).

*SIDT2*, along with others nearby on the same chromosome, has been previously implicated in East Asian GWAS (*Gombojav et al., 2015*). In our results, we found increased predicted gene expression for *SIDT2* to be associated with lower TRIG levels in most models. *SIDT2* has been associated with glucose and lipid metabolism in mice, as *SIDT2* knockout mice have significantly higher serum levels of TRIG than wild-type mice (*Gao et al., 2016*). Since increased gene expression is associated with lower TRIG levels in our cohort for a majority of models, our models concur with the association present within mice. Significance of *SIDT2* for TRIG in LCL was also replicated in GLGC at $P = 1.1 \times 10^{-102}$ (Table 5). The similar effects and significance of *SIDT2* in knockout mice studies, the Cebu cohort analysis, and GLGC PrediXcan results indicate that increased expression of *SIDT2* is associated with decreased TRIG levels. Therefore, variation in the regulation of *SIDT2* may contribute to differences in TRIG levels.

In an attempt to disentangle the multiple genes associated with TRIG at the 11q23.3 locus, we performed backwards elimination modeling. This analysis showed that *BACE1* has the strongest effect at the locus, with *SIDT2*, *APOA1*, and *CEP164* contributing smaller effects.

### rs662799 has a greater impact in East Asian populations

The significant SNPs in the GWAS portion of our study have been previously associated with lipid traits, but these prior studies did not conduct a robust gene-based association modeling (*Hall et al., 2006*; *Buyske et al., 2012*; *Rasmussen-Torvik et al., 2012*; *Wu et al., 2013*). In our PrediXcan analysis, we implicated a new gene (*PAX6*) in Yoruba and additional genes at the chromosome 11 locus in Cebu, including genes with predictive SNPs located hundreds of thousands of base pairs away from their transcription start sites.

rs662799 in particular is a significant SNP in cholesterol GWAS of East Asians (*Teslovich et al., 2010*; *Wu et al., 2013*; *Lin et al., 2016*; *Lu et al., 2016*; *Spracklen et al., 2017*), with its minor allele frequency as high as 0.37 in the 1000 Genomes Japanese (JPT) population, and it is also the top SNP, with a minor allele frequency of 0.245, in our own TRIG GWAS analysis for Cebu (Table 3). rs662799 has a low minor frequency in European and African populations, indicating that even if effects are similar, it has a lower allelic impact in European and African ancestry populations (Fig. 3). In Asian populations, therefore, rs662799 has a more significant genetic impact (*Brown et al., 2016*) due to its higher MAF, causing the difference in phenotypic effects. This demonstrates how prediction models may vary in utility across populations.

## CONCLUSIONS

In this study, we use a transcriptome-informed approach to implicate new genes in lipid traits. Limitations arose from the small sample sizes in desired populations and the lack of population-specific transcriptome prediction models. Other lipid trait GWAS, such as the GLGC, included over 180,000 individuals of European ancestry (*Willer et al., 2013*),

while each population in this study had less than 2,000 individuals. Current GTEx models are based on an 85% European-American and 15% African-American population, which cannot be fully extrapolated to diverse populations. While Yoruba in Ibadan, Nigeria is a HapMap population, and specific reference panels exist for African populations, there is a lack of publicly available data for southeast Asian and Pacific Islander populations, which is an issue due to the rarer variants in more isolated island populations (*Loh et al., 2016*). For example, the reference panel used for imputation, 1000 Genomes Phase 3, contains only Vietnamese, Chinese, and Japanese genotypes for East Asian populations (*Auton et al., 2015*). Without data collection and proper models for non-European populations, there is less potential for accurate implementation of precision medicine. To fully characterize the impact of genetic variation between populations, larger studies in non-European populations are needed.

## ACKNOWLEDGEMENTS

The datasets used for the analyses described in this manuscript were obtained from dbGaP at http://www.ncbi.nlm.nih.gov/gap through accession numbers phs000378.v1.p1 and phs000523.v1.p1. Gene expression prediction models were obtained from PredictDB at http://predictdb.hakyimlab.org/.

### Funding

This work is supported by the National Institutes of Health National Human Genome Research Institute Academic Research Enhancement Award R15 HG009569 (PI: Heather E. Wheeler) and the Loyola University Chicago Carbon Undergraduate Research Fellowship (Angela Andaleon). The funders had no role in study design, data collection and analysis, decision to publish, or preparation of the manuscript.

### Grant Disclosures

The following grant information was disclosed by the authors:
National Institutes of Health National Human Genome Research Institute Academic Research Enhancement Award: R15 HG009569.
Loyola University Chicago Carbon Undergraduate Research Fellowship.

### Competing Interests

The authors declare there are no competing interests.

### Author Contributions

- Angela Andaleon conceived and designed the experiments, performed the experiments, analyzed the data, contributed reagents/materials/analysis tools, wrote the paper, prepared figures and/or tables, reviewed drafts of the paper.
- Lauren S. Mogil analyzed the data, contributed reagents/materials/analysis tools, reviewed drafts of the paper.

- Heather E. Wheeler conceived and designed the experiments, analyzed the data, contributed reagents/materials/analysis tools, wrote the paper, reviewed drafts of the paper.

## Data Availability

The datasets used for the analyses described in this manuscript were obtained from dbGaP at http://www.ncbi.nlm.nih.gov/gap through accession numbers phs000378.v1.p1 and phs000523.v1.p1. Gene expression prediction models were obtained from PredictDB at http://predictdb.hakyimlab.org/. See https://github.com/aandaleon/px_chol for all scripts used in our analyses.

## Supplemental Information

Supplemental information for this article can be found online at http://dx.doi.org/10.7717/peerj.4314#supplemental-information.

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
