# Peer review of "Gene-based association study for lipid traits in diverse cohorts implicates BACE1 and SIDT2 regulation in triglyceride levels"

_PeerJ, doi:10.7717/peerj.4314_

## Round 0.1 · original submission · Major Revisions

My compliments on a well-wrtten and clear paper. As you'll see, the reviewers were generally positive about your manuscript. Reviewer 3, however, who has substantial expertise in this area, raises several substantive issues. Addressing these issues will substantially improve the paper, and in any case they must be addressed before the paper can be accepted.

·

Basic reporting

This manuscript was well-written.

Experimental design

The overall design of this study followed generally acceptable standard in the field. The study questions were defined clearly by the authors. The methods they used are reasonable and appropriate.

Validity of the findings

The authors presented their findings following the generally acceptable standard in the filed of human genetics. All findings are presented clearly with sufficient statistical evidence.

Additional comments

One minor issue is about the comparison between two different array platforms. It is not clear whether the authors took any specific steps to ensure that the same variants (allele, strand direction) were compared. A bit more description in the methods could be helpful.

Reviewer 2 ·

Basic reporting

(1) Clear, unambiguous, professional English language used throughout.
(2) Intro & background to show context. Literature well referenced & relevant.
(3) Structure conforms to PeerJ standards, except for the AKNOWLEDGEMENT section, which should not be used to acknowledge funders according to the PeerJ standards.
(4) Figures are relevant, high quality, well labelled & described.
(5) Although raw data was not supplied, all scripts were shared in GitHub (https://github.com/aandaleon/px_chol).

Experimental design

(1) Original primary research within Scope of the journal.
(2) Research question, which was to clarify the transferability of lipids-associated loci in non-European populations, well defined, relevant & meaningful. It is stated how the research fills an identified knowledge gap.
(3) Rigorous investigation performed to a high technical & ethical standard (GWAS, XPEB, S-PrediXcan).
(4) Methods described with sufficient detail & information to replicate.

Validity of the findings

(1) Data is robust, statistically sound, & controlled.
(2) Conclusions are well stated, linked to original research question & limited to supporting results.

Additional comments

The authors performed GWAS in a Nigerian and a Filipino cohorts, respectively, and XPEB analysis by combining these non-European populations with previously published GLGC GWAS summary statistics in European populations, as well as gene-based analysis through S-PrediXcan. Considering the relatively small sample size of these two non-European populations, it comes as no surprise that no novel loci were identified in the GWAS or the XPEB analysis. Previously reported lipids-associated loci were confirmed in the current study, and novel genes (BACE1 and SIDT2) near known SNP (rs662799) were identified using S-PRediXcan, providing new evidence of the possible underlying mechanisms. The manuscript is well organized and the statistical methods were appropriately implemented.
A few minor comments for the authors.
(1) For the XPEB package, it took SNPs and P values as input as well as the sample size of the included ethnic groups. According to the package manual, SNPs with variation >10% of the median sample size is not acceptable. Roughly half of the GLGC participants were genotyped using common Affymetrix or Illumina GWAS arrays while another half of them were genotyped using Metabochips which included far less SNPs. As a result, a few SNPs in the GLGC dataset showed sample size >10% deviation from the median sample size. Were these SNPs before running XPEB? For the SNPs that were missing in the base population (i.e. GLGC), the package manual provided methods to do imputation. Were the missing SNPs imputed or simply exclude from the analysis?
(2) For Table 3 and Table 4, I recommend add a column showing SNP position in addition to chromosome.
(3) In Supplemental Table 1, results in three different papers were compared. What’s the different between these three datasets?
(4) For Supplemental Table 2, some notes may be needed to explain the meaning of the column names, and also, please be more accurate about the age of the Yoruba study (70-?).
(5) L184-185, six new SNPs for HDL and five new SNPs for LDL had FDR<0.05, at the same loci as the earlier significant SNPs (Supplemental Table 3). This sentence and the results in Supplemental Table 3 were confusing. ‘The earlier significant SNPs’ may refer to the significant findings in the GWAS analysis, which were CETP on chr16 for HDL and APOE on chr19 for LDL. However, there was a significant SNP at PCSK9 on chr1 with FDR<0.05 in Supplemental Table 3. Please rephrase this sentence and correctly describe the results.
(6) For Supplemental Table 4, I recommend re-sort the SNPs first by locus, then by FDR values.

Reviewer 3 ·

Basic reporting

This manuscript is clear, concise, and professionally written. I found the methodology and approach each to interpret.

The introduction was thorough and explained the rationale behind the study.

To the best of my knowledge, basic reporting guidelines were met.

Experimental design

I have some concerns with the experimental design of this study:

1. The authors state that S-PrediXcan/MetaXcan was used on summary statistics from the CEBU and Yoruba cohorts to identify gene-based associations. I do not understand why a summary-statistic based approach was used when raw data is available. Although MetaXcan and PrediXcan are equivalent when using European data, my understanding is that this is not true when using non-European cohorts. I would recommend that authors either a) re-run genic association analyses based on raw data; or b) calculate LD Matrices from the CEBU and YORUBA populations for use with MetaXcan; or c) provide evidence that the two approaches are analogous even when applied to non-European data.

2. I am concerned about the use of the GLGC cohort as a replication data set. The introduction of the manuscript states that the samples includes "7,898 non-European-ancestry individuals, including the children of those in the Cebu cohort". How did the authors account for this potentially substantial relatedness within the cohorts? In particular, I note that the GLGC results have significant overlap with the Cebu PrediXcan results- how was relatedness accounted for in this replication analysis?

Validity of the findings

I have some concern with the validity of the findings in this study.

1. I am concerned that the prediXcan/metaXcan analyses do not include sufficient correction for multiple testing. Currently, authors correct for the number of genes within each tissue, but do not take into account that 44 tissues are tested. My rough estimate (44 tissues * ~5,000 genes/tissue) gives a stricter significance threshold of 2.2e-07. According to this estimate, no genes are significant in the Yoruba population, and <5 in Cebu.
While this correction may be overly stringent given eQTL correlation between tissues, I feel that this is worth noting as a caveat to your results. Perhaps an alternative, less conservative approach, would be to apply a study-wide 5% FDR correction.

2. The authors identify multiple genome-wide significant associations in both the Yoruban and Cebu SNP-based GWAS, which cluster into specific loci. In my opinion, it would be advisable to carry out some type of conditional analysis to identify lead signals in these loci.

For example, for the Cebu cohort, authors list the top 10 significant SNPs in Table 4- these are all in the same locus (Figure 2), and are all in fairly high LD (>0.6). It therefore seems uninformative to list all these SNPs - instead, why not use GCTA-Cojo, FINEMAP, or similar, to identify the lead SNP (s) in this and other loci, and list those?

The same is true for the S-PrediXacn results (eg, presented in Table 6). Authors identify multiple associations in 11q23.3, across multiple tissues (and presumably there are multiple sub-threshold associations in this region also). It would be interesting to see the correlations of GREX between these genes in these tissues, and perhaps to carry out a conditional analysis of these associations.

---

## Round 0.2 · accepted · Accept

Thank you for your substantive responses and changes.

Reviewer 2 ·

Basic reporting

no comment

Experimental design

no comment

Validity of the findings

no comment

Additional comments

The authors have addressed the issues, and I have no further comments.